# Effects of Olive Oil and Its Components on Intestinal Inflammation and Inflammatory Bowel Disease

**DOI:** 10.3390/nu14040757

**Published:** 2022-02-11

**Authors:** Josip Vrdoljak, Marko Kumric, Marino Vilovic, Dinko Martinovic, Iris Jeroncic Tomic, Mladen Krnic, Tina Ticinovic Kurir, Josko Bozic

**Affiliations:** 1Department of Pathophysiology, University of Split School of Medicine, 21000 Split, Croatia; josip.vrdoljak@mefst.hr (J.V.); marko.kumric@mefst.hr (M.K.); marino.vilovic@mefst.hr (M.V.); dinko.martinovic@mefst.hr (D.M.); mkrnic@mefst.hr (M.K.); tticinov@mefst.hr (T.T.K.); 2Department of Public Health, University of Split School of Medicine, 21000 Split, Croatia; iris.jeroncic@mefst.hr; 3Department of Endocrinology, Diabetes and Metabolic Diseases, University Hospital of Split, 21000 Split, Croatia

**Keywords:** olive oil, inflammatory bowel disease, Mediterranean diet, polyphenols

## Abstract

With the rising global burden of inflammatory bowel disease (IBD) and the rising costs of novel biological drugs, there is an increasing need for dietary approaches and functional foods that could modulate the course of IBD. The Mediterranean diet has proven to be efficacious in managing chronic inflammatory diseases, and recent studies have also shown its benefits in the setting of IBD. Since olive oil and its compounds have been shown to provide a considerable anti-inflammatory effect, in this review, we aim to discuss the latest evidence concerning the impact of olive oil and its bioactive compounds on IBD. Numerous preclinical studies have exhibited solid evidence on the mechanisms by which polyphenol-rich extra-virgin olive oil (EVOO) or specific polyphenols like hydroxytyrosol (HT) provide their anti-inflammatory, antioxidative, antitumour, and microbiota-modulation effects. Accordingly, several human studies that explored the effects of olive oil on patients with IBD further confirmed the evidence brought forward by preclinical studies. Nevertheless, there is a need for larger-scale, multicentric, randomized control trials that would finally elucidate olive oil’s level of efficacy in modulating the course of IBD.

## 1. Introduction

Inflammatory bowel disease (IBD) is a chronic, immune-mediated, multifactorial disease of the gastrointestinal (GI) tract, with complex aetiology and pathophysiology [1]. The two main varieties of IBD are ulcerative colitis (UC), which affects the large intestine, and Crohn’s disease (CD), which can affect any part of the GI tract [1]. Global IBD prevalence is rising, with IBD incidence mainly increasing in developing nations [2]. Although the incidence in westernized regions is stabilizing and even decreasing, the prevalence continues to grow because of a younger age of disease onset and better optimized treatment strategies with a subsequent reduction in mortality [3]. The current understanding of IBD pathophysiology is a complex interaction of susceptible genetics, environmental factors (dietary elements, stress), and gut microbiota, in which the immune response to dietary factors and intestinal microbes is disrupted [4,5,6]. The contemporary IBD treatment is comprised of anti-inflammatory drugs (salicylates, corticosteroids), immunomodulating drugs (corticosteroids, azathioprine, 6-mercaptopurine, etc.), and biologic agents (anti-TNF-α, cytokine-targeted therapy, anti-integrin), as well as surgical procedures that are mostly reserved for complications and drug-refractory disease [7,8]. In addition, some dietary treatments like the Exclusive Enteral Nutrition, a specialised liquid nutritional formula, are being used as a first-line therapy for CD [9,10]. Moreover, there is a growing interest in specific nutritional patterns that could potentially modulate the course of IBD, such as the Paleolithic diet, the Mediterranean diet (MedDiet), the low-fermentable oligosaccharides, disaccharides, monosaccharides and polyols (FODMAP) diet, the specific carbohydrate-based diet, the gluten-free diet, and the Groningen anti-inflammatory diet [11,12,13,14].

The MedDiet is a traditional dietary pattern that originated in the olive-tree-bearing regions of the Mediterranean basin. It is defined by high consumption of fruits and vegetables, legumes, nuts, olive oil, unprocessed cereals, fish, and other seafood; the moderate consumption of aged cheese and red wine; and the low intake of meat and dairy products [15]. In recent times, the MedDiet and its specific components are increasingly being researched and used to prevent and manage chronic non-communicable diseases [16,17,18]. In addition, there is compelling new evidence about the efficacy of various nutraceuticals, such as the polyphenols from natural foods like olive oil and wine, in modulating IBD [19,20]. Olive oil, and especially extra virgin olive oil (EVOO), with its high content of mono-unsaturated fatty acids (MUFAs), tocopherols, and polyphenols, is thought to be one of the main bioactive compounds found in the MedDiet [21].

Therefore, two authors (T.T.K. and M.Kr.) independently performed a literature search of three electronic databases (PubMed, Scopus, and Web of Science), with the following search terms used: Olive oil/Extra-virgin olive oil/EVOO AND polyphenols/phenols/biophenols AND intestinal inflammation/Inflammatory bowel disease/IBD/ulcerative colitis/Crohn’s disease. Studies published before 2000 and studies written in a language other than English were excluded from the analysis. For additional studies, we further examined reference lists of relevant research papers. Manuscript screening was performed by checking the title and abstract or reading the full text. We included all the in vitro, animal, and human studies relevant to this paper’s scope.

Hence, this comprehensive review will discuss the current research regarding the effects of olive oil and its compounds on IBD. The composition of olive oil and elucidation of its impact on IBD pathophysiology will be thoroughly described by reviewing the evidence of in vitro, animal, and human studies.

## 2. Olive Oil and IBD

### 2.1. Olive Oil Composition

The benefits of olive oil are traditionally attributed to its high MUFA content, which represents 80% of olive oil’s total lipid composition [22]. Olive oil is primarily made of triacylglycerols (99%) and secondarily of free fatty acids, mono- and diacylglycerols, and a range of other compounds like hydrocarbons, sterols, tocopherols, aliphatic alcohols, and pigments. In addition, an array of phenolic and volatile compounds is also present [23]. Fatty acid representatives in olive oil are palmitic, palmitoleic, stearic, oleic, linoleic, and linolenic acids, while eicosanoic, myristic, and heptadecanoic acids are found in trace amounts [23]. Moreover, the fatty acid compositions often differ depending on the area of production, its latitude, climate, fruit variety, and the stage of maturation [23]. For example, Greek, Italian, and Spanish olive oils are generally low in palmitic and linoleic acids, while they have a high oleic acid content [23]. Considering the phenolic content, olive oil contains hydroxytyrosol (HT) and tyrosol (Tyr), caffeic acid, p-coumaric acid, vanillic acid, flavones (apigenin, luteolin, rutin), as well as secoiridoids (oleuropein, ligstroside, oleacein) [22,24].

The differences between EVOO and other more refined types of olive oil are due to environmental conditions, extraction, and harvesting handlings. Most notably, the extraction procedure has the most considerable influence on the concentration of phenolic compounds. EVOO is produced by cold pressing and processing olives, by which the minor phenolic compounds are preserved. While refined olive oil and EVOO have similar amounts of MUFA, refined oil has a considerably lower phenolic content and hence fewer beneficial effects on health [22]. The main polyphenol in EVOO is HT, which has one of the highest antioxidant capacities of all-natural compounds and has shown antiatherogenic, anticancer, antidiabetic, and neuroprotective effects [25]. Similarly, oleuropein, which is a precursor to HT, exerted anti-inflammatory activity and promoted nitric oxide production, with additional evidence of cardioprotective, neuroprotective, anticancer, and lipid-regulating activity [25,26]. Another polyphenol, oleocanthal, exerts similar health benefits, with anti-inflammatory properties due to NSAID-like mechanisms of action [27]. The reductions in cyclooxygenase (COX)-2, metalloprotease, and interleukin 6 (IL6) are thought to be the leading mechanisms by which olive oil and its biophenols exert anti-inflammatory activity [20,28]. Hence, the current evidence on the anti-inflammatory properties of olive oil polyphenols suggests a potential therapeutic benefit for chronic inflammatory diseases such as IBD.

### 2.2. Olive Oil and IBD

A rising number of research groups is working to determine the potential positive effects of olive oil and its components on IBD. Olive oil polyphenols like HT and Tyr are readily absorbed in the intestine, but have low bioavailability due to the first pass metabolism and the formation of sulphate and glucuronide conjugates [29,30,31]. It was estimated how a standard dose of olive oil found in the MedDiet (50 g) provides about 2 mg of HT-equivalents per day, where the plasma concentration of olive oil phenols resulting from such an intake is at most 0.06 μmol/L [32]. Therefore, the amount of polyphenols in the recommended daily dose of olive oil is not sufficient to achieve all the beneficial effects observed in preclinical studies. However, polyphenols that are not absorbed in the small intestine arrive in the colon, where the gut microbiota can subsequently metabolize them. In line with this, studies have shown that molecules such as oleuropein reach the large intestine as an unmodified compound, and the human colonic microbiota can catabolize oleuropein into HT [33]. Hence, as a higher content of these bioactive polyphenols is present in the GI tract, we could expect that they will exhibit a larger local effect and therefore provide notable benefits in combatting intestinal inflammation, such as that present in IBD.

#### 2.2.1. The Evidence In Vitro

An abnormal immune response of the gut, which leads to an increased release of pro-inflammatory cytokines, along with an increase in reactive oxygen species (ROS) creation, represents a crucial event in IBD pathophysiology. Some of the main contributors in this cascade are TNF-α, COX-2, IL-8, NF-κB, and iNOS [4,34].

Olive oil biophenols, with their high gut concentrations, can exert a direct antioxidant effect, while also modulating the intestinal epithelial homeostasis by positively affecting inflammation and the gut microbiota [35]. In a study by Cardeno et al., peritoneal macrophages isolated from mice were stimulated with LPS and then treated with the unsaponifiable fraction (UF) of EVOO. UF showed anti-inflammatory and antioxidant effects by inhibiting the LPS-induced intracellular ROS and nitrite production [36]. In addition, UF decreased the COX-2 and iNOS protein expression by down-regulating the NFκB signal pathway and MAPK phosphorylation [36]. Interestingly, another study by the same authors investigated the in vitro effects of EVOO’s UF on intestinal T cells from IBD patients and healthy subjects [37]. The presence of UF promoted apoptosis and reduced the activation of T cells isolated from IBD patients. The frequency of CD69(+) and CD25(+) T cells was decreased, and IFN-γ secretion was reduced [37].

Additionally, another study that researched the effects of HT on murine macrophages exposed to LPS showed how HT inhibited the production of NO and PGE2 while also reducing the secretion of other pro-inflammatory cytokines and chemokines like IL-6, TNF-α, and CXCL10/IP-10 [38]. These effects were in part mediated by inhibiting the NFκB pathway, therefore elucidating the molecular basis for HT’s anti-inflammatory activity [38].

Furthermore, in the study by Serra et al., the researchers evaluated the ability of an EVOO phenolic extract to counter the pro-inflammatory/pro-oxidant effects of dietary oxysterols in differentiated enterocyte-like cells of a human adenocarcinoma cell line (Caco-2) [39]. Pre-treatment with EVOO phenolic extract counteracted the increases in IL-8, IL-6, and iNOS by modulating the MAPK-NF-kB pathway [39]. Similarly, another study by the same author elucidated the effects of EVOO phenolic extract in ex vivo human immune cells treated with oxysterols. While the oxysterol treatment increased the production of pro-inflammatory cytokines and ROS species, the addition of EVOO phenols significantly reduced cytokine secretion and inhibited ROS production and MAPK phosphorylation [40]. Interestingly, analogous findings were reported in studies exploring the effects of wine polyphenols in modulating gut inflammation [19]. Namely, a study by Guina et al. demonstrated how wine polyphenols counteracted the aforementioned effect of oxysterols by inhibiting the NOX1/p38 MAPK/NFκB signalling axis [41]. This shows how polyphenols from two major parts of the MedDiet, olive oil and wine, provide similar beneficial anti-inflammatory, antioxidative, and immune-modulatory effects.

Other protective effects of EVOO were also demonstrated in an in vitro model of alternariol (mycotoxin)-induced cytotoxicity in Caco-2 cells. Alternariol has cytotoxic effects and leads to increased ROS production, while the addition of EVOO extract had a significant cytoprotective and antioxidative effect [42]. In a study by Muto et al., the researchers investigated the ability of EVOO phenolic extract to modulate the inflammatory response in undifferentiated and differentiated Caco-2 cells that were challenged with LPS or IL-1β in order to mimic intestinal inflammation [43]. In Caco-2 cells treated with LPS (reflecting early phase of inflammation), the EVOO phenolic extract prevented IL-8 expression and secretion. On the other hand, in the cells treated with IL-1β (reflecting intermediate phase of inflammation), EVOO inhibited IL-8 promoter activity but led to an increase in IL-8 mRNA stability and protein expression, with the latter mechanism prevailing over the first one [43]. Thus, it was concluded that EVOO has a complex role in the regulation of intestinal inflammation by both transcriptional and posttranscriptional mechanisms.

The major polyphenols found in olive oil, HT and Tyr, are both significantly metabolised in the intestine, and their glucuronide and sulphate metabolites concentrate at the intestinal level [29]. Serelli et al. explored the effects of HT and Tyr metabolites in Caco-2 cells treated with LPS [44]. They showed how HT and Tyr metabolites inhibited iNOS expression and were effective in the inhibition of IĸBα degradation, therefore leading to less NFκB activity [44]. Moreover, in another study that investigated the effects of olive polyphenols on intestinal cells, the researchers showed how polyphenols sourced from olive pomace (usually a by-product of the olive oil industry) also have notable anti-inflammatory effects [45]. The Caco-2 cells were treated with IL-1β, and the addition of aqueous extract of olive pomace led to a significant reduction in IL-8 secretion in both the basal and inflamed condition [45]. In addition, a study that investigated the effects of EVOO phenolic extracts in the setting of Caco-2 cells treated with tert-butyl hydroperoxide (TBH) or a mixture of oxysterols showed how a preincubation with the phenolic extracts significantly reduces oxidative modifications [46]. Similar antioxidative effects of EVOO were produced in another study, where the researchers investigated the antioxidant properties of fractions obtained from six Spanish monovarietal EVOOs [47]. They evaluated the antioxidant activity of EVOO’s bioaccessible fractions (BF), obtained after in vitro digestion. The in vitro digestion process increased the total phenolic count and antioxidant activity, while all the studied EVOO varieties showed a significant reduction in ROS production when incubated with Caco-2 cells [47].

Olive oil and its phenolic compounds have also proven their benefits in modulating colorectal carcinogenesis. Gill et al. explored the effects of EVOO phenols on a series of in vitro systems that modelled important stages of colon carcinogenesis [48]. HT-29 cells were treated with hydrogen peroxide, and a significant anti-genotoxic effect was observed when the cells were pre-treated with olive oil phenols. The olive oil phenols also significantly enhanced the barrier function of Caco-2 cells, whereas they reduced HT115 cell invasion and attachment, thereby demonstrating how they can inhibit several stages of in vitro colon carcinogenesis [48]. Other studies have additionally displayed the anticancer effects of HT and its colonic metabolites, namely through cell cycle arrest and apoptosis [49,50].

After an additional thorough search for studies that are directly related to IBD patients, to the best of our knowledge, there are two studies that investigated the ex vivo organ cultures of mucosal explants from UC and CD patients [20]. In the first study, biopsies were obtained during colonoscopy from 14 patients with active UC and were immediately placed in an organ culture chamber and challenged with LPS from Escherichia coli, in the presence or absence of oleuropein (OLE) [51]. The expression of IL-17 and COX-2 were significantly lower in samples treated with OLE. Furthermore, the OLE-treated colonic samples showed signs of mucosal healing, with reduced infiltration of CD3, CD4, and CD20 cells [51]. In the second study, the researchers first studied the anti-inflammatory effects of olive leaf extract (0.5–25 mg/kg), which contained more than 80% OLE, in two mice models of colitis (DSS and DNBS) [52]. Later on, they also evaluated the immunomodulatory effects of an olive leaf extract in ex vivo colon cultures from CD patients. In both colitis models, the olive extract reduced the expression of proinflammatory mediators (IL-1β, TNF-α, and iNOS) and improved the intestinal epithelial barrier by restoring the expression of ZO-1, MUC-2, and TFF-3. These effects were confirmed in the ex vivo model from CD patients, where the olive extract reduced the production of proinflammatory mediators (IL-1β, IL-6, IL-8, and TNF-α) [52].

Overall, the evidence produced through these in vitro studies shows a clear effect of olive oil phenols on major inflammatory pathways (Figure 1). The commonly observed effects were inhibition of p38/MAPK, NFκB, and iNOS. Furthermore, a significant antioxidative and anticancer ability was also observed. Therefore, in vitro evidence reveals how olive oil and its phenols can modulate several key points in IBD pathophysiology (Table 1).

#### 2.2.2. Evidence from Animal Studies

The evidence gathered from the in vitro studies was further expanded and confirmed on animal models, mostly on mice and rat models with dextran sulphate sodium (DSS)-induced colitis. In one study that combined in vitro and animal colitis models, the authors investigated the influence of medium-chain triglycerides (MCT), olive oil components (oleic acid and HT), and fish oil (n-3 fatty acids) on colitis modulation [53]. First, they demonstrated how oleic acid and HT reduce the t-butyl hydroperoxide-induced damage on Caco-2 cells, while the MCT exaggerated the damage. Further, they showed how a combined treatment of cells with eicosapentaenoic acid, docosahexaenoic acid, oleic acid, and HT had a synergistic effect, leading to larger protective effects and lower IL-8 production. Later, they confirmed these findings in a DSS-colitis rat model, where a fish oil and olive oil combination led to a significant attenuation of DSS-induced alterations. On the other hand, the MCT group exhibited more disease activity, with higher levels of inflammatory cytokines in the colon [53].

Sanchez-Fidalgo et al. studied the influence of an EVOO diet enriched with HT in a chronic mouse DSS-colitis model [54]. The mice were randomized into three groups: standard diet, EVOO, and EVOO enriched with HT. Mice that were fed a diet with EVOO had considerably better histological signs, improved disease activity index, as well as a 50% reduction in mortality caused by DSS. Additionally, HT supplementation produced better results than EVOO alone, with a better histological index and a larger reduction in iNOS levels. In both EVOO and EVOO + HT groups, the levels of TNF-α, iNOS, and p38MAPK activation were reduced, whereas the IL-10 (anti-inflammatory cytokine) was significantly increased [54].

Similarly, another study on chronic colitis in mice, which analysed the effects of EVOO enriched with polyphenols, reported similar results; the disease activity index, cell proliferation, as well as MCP-1, TNF-α, COX-2 and iNOS expression levels were all significantly reduced [55]. Notably, the mechanisms behind the anti-inflammatory effects exhibited by the treatment group were elucidated by a marked down-regulation in JNK phosphorylation, with IκBα and PPARγ up-regulation [55]. Further investigation by the same research group analysed the effects of EVOO’s unsaponifiable fraction (UF) on the acute ulcerative colitis model in mice [56]. The mice were randomized into three groups of 20. One group of mice was fed sunflower oil (SD), while the others were fed with an EVOO diet or a UF-enriched SD at 5% oil (SD+UF). The histological score and disease activity index were significantly improved in both EVOO and SD+UF dietary groups versus the SD group. Additionally, both groups achieved a significant reduction in MCP-1 and TNF-α levels, iNOS, COX-2, and p38 MAPK exhibited lower expression and activity, and IκB expression was augmented [56].

Likewise, in a study by Takashima et al., the authors demonstrated the benefits of an EVOO diet in a rat model of chronic DSS-induced colitis [57]. The rats were divided into three groups: (1) control group (no DSS), (2) DSS group on a standard diet, (3) DSS group on EVOO diet. The EVOO diet significantly attenuated the expression of inflammatory factors (STAT3, pSTAT3, COX-2, and iNOS), and it also decreased the levels of cellular proliferation (measured by PCNA expression). Additionally, the EVOO diet increased cleaved caspase-3 levels, leading to a recovery of apoptosis [57].

Correspondingly, Carrielo et al. demonstrated the positive effects of EVOO from four Apulian cultivars on a mouse model of DSS-induced colitis [58]. Administration of EVOO resulted in improved intestinal morphology, less body weight loss, and reduced rectal bleeding and IL-1β, TGFβ, IL-6 gene expression levels [58]. Furthermore, in a study that focused on the effects of an EVOO-enriched diet in the setting of DSS-colitis associated colon carcinogenesis in mice, the researchers found that EVOO-fed mice exhibited less incidence and multiplicity of tumours than those that were fed with sunflower oil [59]. On the other hand, Nascimento et al. studied the impacts of EVOO and flaxseed oil (FO) in a DSS-induced mice model of UC and found no notable benefits of the aforementioned diets [60]. In this study, eighty 8-week-old C57BL/6J mice were placed into four groups: Control, 10% EVOO, 10% FO, and 5% EVOO + 5% FO, with the oils supplementing the AIN-93M diet. Interestingly, there were no differences in disease activity index (DAI), histopathological score, interleukin (IL)-1β, and iNOS between the DSS and treatment groups, and the EVOO + FO group even displayed a significant increase in TNF-α levels. Notably, only IL-6 levels were significantly decreased in the EVOO + FO group [60].

Oleuropein (OLE), a notable anti-inflammatory and antioxidant polyphenol from olive oil, was investigated in an interesting study by Huguet-Casquero et al. [61]. They researched the effects of OLE by itself (OLEsus), and OLE loaded with nanostructured lipid carriers (NLCs), which have been shown to accumulate in inflamed colonic mucosa. The compound was first examined in vitro on activated macrophages (J774), where NLC-OLE was shown to be more efficacious in decreasing the TNF-α secretion and intracellular ROS levels. Further, in a murine model of acute colitis, both the OLEsus and NLC-OLE groups showed a significant decrease in MPO activity as well as TNF-α and IL-6 concentration, but with no differences between the two groups. Strikingly, only the NLC-OLE group produced a significant decrease in total ROS generation [61]. Moreover, as was reviewed before, an intrarectal administration of an aqueous solution containing olive oil and HT reduced the severity of inflammatory damage in a Wistar rat model of TNBS-induced colitis, with notable safety characteristics, since the treatment did not show any adverse effect in the animals [20,62].

In an older study, Camuesco et al. examined the effects of dietary olive oil supplemented with fish oil (EPA, DHA) in a rat model of DSS-induced colitis [63]. Colitic rats fed with an olive-oil-based diet had a lower colonic inflammatory response than those fed a soybean oil diet. At the same time, this effect was also increased with dietary (n-3) PUFA supplementation. A restoration of colonic glutathione levels was observed in the olive oil group, along with a decrease in iNOS expression. Moreover, the addition of (n-3) PUFA into the olive oil diet produced a further significant decrease in TNF-α and LTB-4 levels [63].

Interestingly, one study that investigated EVOO in HLA-B27 transgenic rats with intestinal inflammation found that EVOO did not influence disease signs like diarrhoea, myeloperoxidase activity, and mucosal injury [64]. Nevertheless, EVOO did significantly reduce TNF-α gene expression in the colon mucosa and decreased total cholesterol blood levels compared to HLA-B27 rats fed with 10% corn oil. The authors argue that the differences from previous studies that examined EVOO and intestinal inflammation probably arose from the varied experimental model (genetically driven chronic colitis vs. DSS-induced). Likewise, the daily dose of HT was only 90 µg/kg bw, which is much lower than the amount (approximately 400 µg/kg/bw) used in a study by Sanchez-Fidalgo et al. [54,64].

Studies on EVOO and its components have also indicated that EVOO/polyphenols promote the health of the gut microbiota, acting like a prebiotic to beneficial bacteria like Lactobacillus and Bifidobacterium, which both utilize oleuropein as a carbon source [35,65,66]. Others have shown how EVOO consumption raised the concentrations of Bacteroides spp., especially B. fragilis, while hindering the growth of Proteobacteria, Firmicutes, Deferribacteres, and Rikenella [65,67,68,69].

Overall, the animal studies mostly confirm the evidence from in vitro studies and display how EVOO and its primary polyphenols like oleuropein and hydroxytyrosol positively affect intestinal inflammation. The common effects can be summed up in the following categories: (1) suppression of inflammatory pathways (decrease in(↓) COX, ↓ iNOS, ↓ p38MAPK) and cytokines (↓ TNF-α, ↓ IL-6), (2) improvement in disease activity index (DAI), (3) improved histopathological signs, and (4) healthier gut microbiota biodiversity. In addition, a recent systematic review and metanalysis that investigated olive-based interventions in murine models concluded how statistically significant outcomes, with moderate-to-large effect sizes, were found for a milder disease expression, better weight maintenance, and reduced rectal bleeding [70].

However, not all studies have reproduced these benefits. The study from Nascimento et al. found EVOO did not significantly impact disease prevention and management [60]. Likewise, in a study by Bigagli et al., EVOO did not influence disease signs like diarrhoea, myeloperoxidase activity, and mucosal injury [64]. These differences could arise from varied compositions of the olive oil used (varied polyphenol content). In addition, some studies used EVOO enriched with HT and EVOO augmented with UF, therefore boosting the polyphenolic content, which could be the cause of efficacy [54,55,56]. Thus, we have to be very cautious in translating the presented evidence into a real-life clinical setting. An overview of the animal studies that investigated the effect of the olive oil polyphenols on intestinal inflammation can be seen in Table 2.

#### 2.2.3. Evidence from Human Studies

It has been demonstrated how the MedDiet promotes a healthy intestinal microbiota and modulates the pathogenesis and course of IBD [70,71,72,73,74]. On the other hand, clinical trials concerning specific parts of the MedDiet, like olive oil, and patients with IBD are very scarce.

Morvaridi et al. performed a randomized crossover clinical trial that investigated the effect of EVOO and canola oil (CO) in UC patients [75]. Of the forty patients eligible for the study, thirty-two patients completed both intervention rounds. The patients consumed 50 mL of either EVOO or CO daily for 20 days, followed by a 14 days washout period and another 20 days of alternate intervention. There was a significant decrease in erythrocyte sedimentation rate (*p* = 0.030) and high-sensitivity C-reactive protein (*p* < 0.001) after EVOO consumption. Furthermore, EVOO consumption significantly reduced symptoms like bloating, constipation, faecal urgency, and incomplete defecation, with a significantly reduced final gastrointestinal symptom rating scale (GSRS) (*p* < 0.05) [75].

In a recent study performed on ten hypercholesterolemic patients, Martín-Peláez et al. investigated the benefits of olive oil phenolic compounds with regards to intestinal immunity [76]. This was a randomized controlled, double-blind crossover trial, where during three weeks, the participants ingested 25 mL/day of three raw virgin Olive Oils (OOs) differing in their polyphenol (PC) concentration and origin, after which there was a two-week wash-out period. Interestingly, the ingestion of PC-enriched OO containing 500 mg PC/kg increased the proportions of IgA-coated bacteria and led to a small but statistically significant increase in the plasma levels of CRP (1.4 (0.33, 5.71) to 2.6 (0.51, 13.02)) [76]. While this confirms the ability of olive oil polyphenols to stimulate the intestinal immune system, it is questionable whether these effects would be detrimental in the setting of IBD, where the intestinal immunity is already overstimulated.

Another randomized controlled trial from the same authors evaluated the changes in faecal microbiota after three weeks of ingestion of 25 mL/day of phenolic-enriched virgin olive oil (VOO) [77]. The results showed a significant increase in beneficial Bifidobacterium, together with an increase in faecal HT and dihydroxyphenylacetic acids [77].

While olive oil has proven to be an important part of the MedDiet’s efficacy, there are studies that show the influence of the MedDiet as a whole on patients with IBD. For instance, Chicco et al. conducted a study on 142 IBD patients (84 UC and 58 CD) who followed a MedDiet for six months [78]. Adherence to the MedDiet improved BMI and waist circumference, while it also reduced inflammatory biomarkers, liver steatosis, and the number of patients with active disease [78].

In a prospective cohort of 25,639 people recruited from the EPIC study (European Prospective Investigation into Diet and Cancer), it was shown (after 7–11 years of follow-up), that oleic acid use represents a protective factor for UC incidence, as the highest tertile of oleic acid intake was inversely associated with UC incidence (OR 0.03, 95% CI 0.002–0.56) [79]. Thus, the study exhibited additional benefits of olive oil in the primary prevention of UC.

The promising results from the studies mentioned above reproduce some of the effects seen in animal and in vitro studies, like anti-inflammatory effects, symptom reduction, and positive microbiota changes [75,76,77]. However, we must be cautious in translating these findings to practice. Namely, the amounts of olive oil compounds used in clinical studies usually overestimate the amount found when olive oil is used as a nutrient. Finally, the pharmacodynamic and pharmacokinetic properties of the aforementioned compounds have yet to be established. Nevertheless, as a substantial amount of accumulated data suggests the safety and, to some extent, the beneficial effects of olive oil, it is well-justified to perform randomized trials that would confirm these effects and establish the appropriate dosage/type of olive oil in IBD treatment. An especially interesting aspect would be to determine the benefits, if there are any, of olive oil in the primary and secondary prevention of cardiovascular repercussions, as IBD patients are burdened by increased risk for atherosclerotic cardiovascular disease and venous thromboembolism [80,81].

## 3. Conclusions and Future Perspectives

This review summarised the current evidence concerning the effects of olive oil and its specific polyphenols on IBD. Numerous in vitro and animal studies provide solid evidence of the mechanisms by which olive oil and polyphenols like oleuropein and HT exert their antioxidative, anti-inflammatory, immunomodulatory, and anti-tumour effects. A scarcity of clinical studies performed on patients with IBD showed promising results, with excellent patient adherence and safety profiles [75]. In addition to the anti-inflammatory effects, the interventional study by Chicco et al. further extends the benefit spectrum of the MedDiet in IBD patients (↓ liver steatosis, ↓ patients with active disease, increase in(↑) QoL). It would be interesting to see if similar interventional trials, with a focus on olive oil and its polyphenols, could reproduce these results.

All of the accumulated evidence sets the MedDiet, and olive oil specifically, as an excellent dietary intervention that may supplement standard IBD medications and help patients in managing their disease. That being said, we still lack large-scale, multicentric, randomized control trials that would finally elucidate olive oil’s level of efficacy in modulating the course of IBD.

## Figures and Tables

**Figure 1 nutrients-14-00757-f001:**
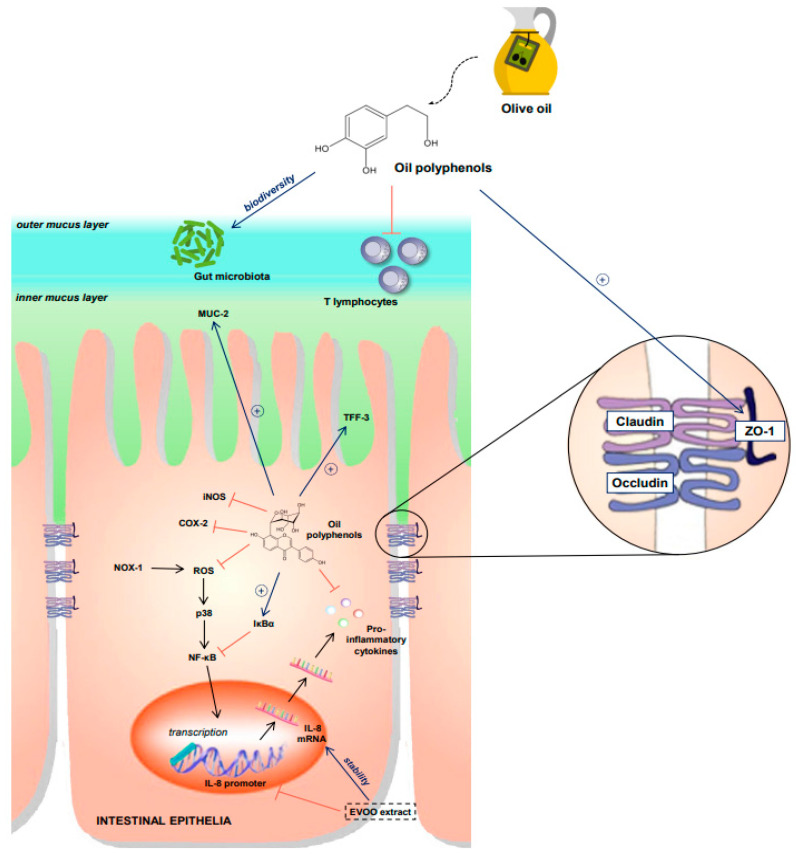
Effects of olive oil polyphenols on intestinal homeostasis as shown by in vitro studies. Abbreviations: ROS: reactive oxygen species, IĸBα: inhibitor of ĸappa Bα, iNOS: inducible nitric-oxide synthetase, IL-8: interleukin 8, COX 2: cyclooxygenase 2, ZO-1: zonula occludes protein 1, MUC-2: mucin 2, TFF-3: intestinal trefoil factor.

**Table 1 nutrients-14-00757-t001:** In vitro studies on olive oil polyphenols and intestinal inflammation.

Study	Cell Type	Intervention	Results
Chiesi et al. [42]	Caco-2 cells stimulated with alternariol	EVOO extract, Oleuropein, Tyrosol	↓ ROS ↓ cytotoxicity
Muto et al. [43]	Caco-2 cells stimulated with LPS or IL-1β	EVOO phenolic extract	↓ IL-8 expression and secretion
Serelli et al. [44]	Caco-2 cells treated with LPS	HT and Tyr metabolites	↓ degradation of IĸBα ↓ iNOS expression
Di Nunzio et al. [45]	Caco-2 cells treated with IL-1β	Polyphenols sourced from olive pomace	↓ IL-8
Incani et al. [46]	Caco-2 cells treated with tert-butyl hydroperoxide (TBH) or a mixture of oxysterols	Preincubation with the phenolic extracts	↓ ROS
Borges et al. [47]	Caco-2 cell cultures	Six Spanish monovarietal EVOOs (EVOOs’ bioaccessible fractions (BF) after in vitro digestion	↑ phenolic count and anti-oxidant activity ↓ ROS
Gill et al. [48]	In vitro model of colon carcinogenesis (HT-29 cells treated with hydrogen peroxide, Caco-2 cells, HT115 cells)	EVOO phenols	anti-genotoxic effect ↑ barrier function in Caco-2 cells ↓ HT115 cell invasion and attachment
Larrussa et al. [51]	Ex vivo organ culture of mucosal explants from UC patients	Oleuropein	↓ COX 2 and IL-17 expression ↓ infiltration of CD3, CD 4 and CD20 cells ↑ mucosal healing
Vezza et al. [52]	Ex vivo colon cultures from CD patients, DSS and DNBS mice colitis models	Olive leaf extract	↓ expression of IL-1β, TNF-α, and iNOS ↑ epithelial barrier (ZO-1, MUC-2, and TFF-3)

Abbreviations: EVOO: extra-virgin olive oil, ROS: reactive oxygen species, IĸBα: inhibitor of ĸappa Bα, iNOS: inducible nitric-oxide synthetase, IL-8: interleukin 8, IL-17: interleukin 17, TNF-α: tumour necrosis factor alpha, COX 2: cyclooxygenase 2, ZO-1: zonula occludes protein 1, MUC-2: mucin 2, TFF-3: intestinal trefoil factor, ↓ = decrease in, ↑ = increase in.

**Table 2 nutrients-14-00757-t002:** Animal studies on olive oil polyphenols and intestinal inflammation.

Study	Cell Type	Intervention	Results
Reddy et al. [53]	DSS-colitis rat model, and Caco-2 cells treated with t-butyl hydroperoxide	Oleic acid and HT, fish oil, MCT	Synergistic anti-inflammatory effect between olive oil PP and fish oil; MCT increased disease activity
Sanchez-Fidalgo et al. [54]	Chronic mouse DSS-colitis model	EVOO diet enriched with HT	↓ disease activity index ↑ histological signs ↓ 50% reduction in mortality ↓ TNF-α, iNOS, and p38MAPK ↑ IL-10
Sanchez-Fidalgo et al. [55]	Chronic mouse DSS-colitis model	EVOO enriched with polyphenols	↓ MCP-1, TNF-α, COX-2 and iNOS expression
Sanchez-Fidalgo et al. [56]	Acute ulcerative colitis model in mice	EVOO’s unsaponifiable fraction (UF)	↓ MCP-1 and TNF-α levels, iNOS, COX-2, and p38MAPK ↑ IκB expression
Takashima et al. [57]	Chronic DSS-induced colitis in rats	EVOO diet (5% of weight)	↓ STAT3, pSTAT3, COX-2 and iNOS ↓ cell proliferation (PCNA) ↑ apoptosis (caspase-3)
Carrielo et al. [58]	Mouse model of DSS-induced colitis	EVOO from 4 Apulian cultivars	↑ intestinal morphology ↓ body-weight loss ↓ rectal bleeding ↓ IL-1β, TGFβ, IL-6 gene expression
Sanchez-Fidalgo et al. [59]	DSS-colitis-associated colon carcinogenesis in mice	EVOO enriched diet	↓ incidence and multiplicity of tumours
Nascimento et al. [60]	Mouse model of DSS-induced colitis	EVOO and flaxseed oil	↓ IL-6 No differences in disease activity index (DAI), histopathological score, interleukin (IL)-1β, and iNOS between the DSS and treatment groups
Huguet-Casquero et al. [61]	Activated macrophages (J774), murine model of acute colitis	Oleuropein (OLE) and OLE loaded with nanostructured lipid carriers	↓ MPO activity, TNF-α and IL-6
Voltes et al. [62]	Wistar rat model of TNBS-induced colitis	Intrarectal administration of aqueous solution containing olive oil and HT	↓ inflammatory infiltrate
Bigagli et al. [64]	Colitis induced in HLA-B27 transgenic rats	AIN-76 diet containing 10% corn oil or extra-virgin olive oil with high (EVOO) or low phenolic content (ROO)	No differences in disease signs like diarrhoea, myeloperoxidase activity, and mucosal injury

Abbreviations: EVOO: extra-virgin olive oil, IL-1B: interleukin 1 beta, TGFB: tumour growth factor B, MPO: myeloperoxidase, MCP-1: monocyte chemoattractant protein-1, IĸB: inhibitor of ĸappa B, COX 2: cyclooxygenase 2, DSS: dextran sulphate sodium, TNBS: trinitrobenzenesulfonic acid, MAPK: mitogen activated protein kinase, TNF-α: tumour necrosis factor alpha, MCT: medium chain triglyceride, HT: hydroxytyrosol, AIN-76: purified rodent diet, STAT3: signal transducer and activator of transcription 3, PCNA: proliferating cellnuclear antigen, ↓ = decrease in, ↑ = increase in.

## Data Availability

Not applicable.

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
