# Peer review of "Effects of Olive Oil and Its Components on Intestinal Inflammation and Inflammatory Bowel Disease"

_nutrients, 2022, doi:10.3390/nu14040757_

Round 1

Reviewer 1 Report

The study on the preventive effect of olive oil on inflammatory bowel disease (IBD) is very interesting. In order to know the preventive effect of olive oil on human IBD, polyphenols which are the active ingredients of olive oil must be absorbed by the human body. Please discuss its effectiveness by adding literature on blood polyphenol levels when ingesting olive oil.

Author Response

Reviewer #1

First of all, thank you for taking the time to thoroughly analyze our study and for giving us valuable suggestions for improving it. Disregarding the results of this revision, we sincerely appreciate your effort!

  1. The study on the preventive effect of olive oil on inflammatory bowel disease (IBD) is very interesting. In order to know the preventive effect of olive oil on human IBD, polyphenols which are the active ingredients of olive oil must be absorbed by the human body. Please discuss its effectiveness by adding literature on blood polyphenol levels when ingesting olive oil.

Dear reviewer, thank you for your important observation.  We have now expanded our manuscript with discussion concerning the polyphenol levels found in the bloodstream upon ingestion of olive oil. namely, although polyphenol GI absorption is relatively scarce, thus limiting the favourable effects of olive oil observed in preclinical studies, relatively high levels of polyphenols and its metabolites are present throughout GI tract, we could expect that they will exhibit a larger local effect, and therefore provide notable benefits in combating intestinal inflammation, such as the one present in IBD.

Reviewer 2 Report

Vrdoljak et al. submitted a review to the journal on effects of olive oil and its components on IBD. The strength is that the authors covered a wide range of experimental models (from in vitro to human), which suggests possible beneficial effects of MedDiet and olive oil. The authors cited a number of recently published articles ( > 2017). Information is well organized in tables and the structure of manuscript is also complementary. However, I have two minor concerns.  

  1. The authors did not clearly explain how literatures reviewed in the manuscript have been searched and selected. This type of paper should include the ‘inclusion and exclusion criteria’ and ‘time range’. Please state these in the introduction. 
  2. An illustration figure(s) that describes the molecular mechanisms (including the major signaling molecules, cytokines involved; based on in vivo and in vitro) by which EVOO and other compounds act on IBD will be very helpful. I suggest the authors include one.

Author Response

Reviewer #2

Vrdoljak et al. submitted a review to the journal on effects of olive oil and its components on IBD. The strength is that the authors covered a wide range of experimental models (from in vitro to human), which suggests possible beneficial effects of MedDiet and olive oil. The authors cited a number of recently published articles ( > 2017). Information is well organized in tables and the structure of manuscript is also complementary. However, I have two minor concerns. 

First of all, thank you for taking the time to thoroughly analyze our study and for giving us valuable suggestions for improving it. Disregarding the results of this revision, we sincerely appreciate your effort!

  1. The authors did not clearly explain how literatures reviewed in the manuscript have been searched and selected. This type of paper should include the ‘inclusion and exclusion criteria’ and ‘time range’. Please state these in the introduction.

Dear reviewer, thank you for your important observation. We have now added a paragraph in which literature search strategy has been elucidated.

  1. An illustration figure(s) that describes the molecular mechanisms (including the major signaling molecules, cytokines involved; based on in vivo and in vitro) by which EVOO and other compounds act on IBD will be very helpful. I suggest the authors include one.

Dear reviewer, thank you for your important observation. We have now added a comprehensive figure concerning the putative pathways by which olive oil compounds may affect GI inflammation in IBD.